# Optimization Issues of a Hammer Mill Working Process Using Statistical Modelling

**Gigel Paraschiv [1], Georgiana Moiceanu [2], Gheorghe Voicu [1,*], Mihai Chitoiu [1], Petru Cardei [3], Mirela Nicoleta Dinca [1] and Paula Tudor [2]**

1   Department of Biotechnical Systems, University POLITEHNICA of Bucharest, 060042 Bucharest, Romania; gigel.paraschiv@upb.ro (G.P.); mihai_chitoiu@yahoo.com (M.C.); mirela.dinca@upb.ro (M.N.D.)
2   Department of Management and Entrepreneurship, University POLITEHNICA of Bucharest, 060042 Bucharest, Romania; georgiana.moiceanu@upb.ro (G.M.); paula.voicu@upb.ro (P.T.)
3   The National Institute of Research—Development for Machines and Installations Designed for Agriculture and Food Industry—INMA Bucharest, 013813 Bucharest, Romania; cardei@inma.ro
*   Correspondence: gheorghe.voicu@upb.ro; Tel.: +40-7247-15585 or +40-7452-01365

**Abstract:** Our paper presents the hammer mill working process optimization problem destined for milling energetic biomass (*Miscanthus Giganteus* and *Salix Viminalis*). For the study, functional and constructive parameters of the hammer mill were taken into consideration in order to reduce the specific energy consumption. The energy consumption dependency on the mill rotor spinning frequency and on the sieve orifices in use, as well as on the material feeding flow, in correlation with the vegetal biomass milling degree was the focus of the analysis. For obtaining this the hammer mill was successively equipped with 4 different types of hammers that grind the energetic biomass, which had a certain humidity content and an initial degree of reduction ratio of the material. In order to start the optimization process of hammer mill working process, 12 parameters were defined. The objective functions which minimize hammer mill energy consumption and maximize the milled material percentage with a certain specific granulation were established. The results obtained can serve as the basis for choosing the optimal working, constructive, and functional parameters of hammer mills in this field, and for a better design of future hammer mills.

**Keywords:** energetic crop biomass; size reduction; hammers mill; energy consumption; process optimization

## 1. Introduction

Lately, the depletion of conventional energy resources and concern for reducing greenhouse gas emissions represent the main argument for the use of renewable resources for energy production. The renewable energy production should be sustainable from an economic and environmental point of view. In this context, woody biomass has become popular as a feedstock for bioenergy generation that is an essential substitute for fossil energy [1,2].

Lignocellulosic biomass is a promising source of energy, available in large quantities which do not compete with food production security and, thus, contributes to environmental sustainability. The use of renewable energy from biomass is one of the most cost-effective and available technologies that can decrease $CO_2$ emissions. The main lignocellulosic biomass resources include agricultural residues, energy crops, forest residues, as well as some of the municipal solid waste [3,4].

Lignocellulosic biomass is mainly composed of lignin, cellulose, and hemicelluloses, but also other elements are found in smaller percentages like other carbohydrates, ash, pectin, and proteins [5]. Due to lignocellulosic biomass recalcitrance which represents biomass resistance to chemical and biological breakdown, its conversion in energy is more difficult than other biomass resources. Therefore, the pretreatment represents an

essential factor in the biorefinery process, the biomass characteristics are improved and higher digestible cellulose for enzymatic hydrolysis is obtained. Lately, different pretreatment techniques were recommended for lignocellulosic biomass in order to remove/alter lignin, hemicellulose, or cellulose structures. The pretreatment methods include physical (mechanical), chemical, and biological but these can be used single or in combinations [6,7].

Mechanical pretreatment is considered a key operation for treating and transforming biomass into biofuel before passing to the subsequent operations [8]. Enzymatic degradation can be improved by milling as it reduces the degree of crystallinity and material size [9]. Size reduction in biomass also increases the bulk density, which improves the flow of biomass during densification. The size reduction methods for biomass comminution include chopping, chipping, hammer milling, crushing, shredding, and grinding [4].

Because of the heterogeneous structure of lignocellulosic biomass, the size reduction by milling process involves several steps, such as: generally coarse milling (from m to cm), intermediate comminution (from cm to 1 mm), fine milling (between 50 and 500 μm), and ultra-fine milling (<20 μm) [10].

Milling can be divided into dry or wet milling and is selected depending on the type of biomass used. Extruders, roller mills, cryogenic mills, and hammer mills are commonly used for dry biomass [7].

The main advantages of milling pretreatment include increasing the hydrolysis rate, does not produce inhibitory or toxic byproducts and can reduce the storage fee of the pulverized feedstock [6,11]. Thus, milling it is a preferred preliminary pretreatment method for a wide variety of lignocellulosic feedstocks [6].

However, the main drawback of the milling pretreatment is its high-energy requirement and the capital cost of mechanical equipment. The specific energy consumption in relation to final particle size is one of the most important economical parameters in the choice of milling equipment [11].

Different parameters are responsible for the energy consumption during the milling process, such as the mill motor speed, the material feeding flow, the initial properties of biomass, the biomass particles size before and after milling, the moisture content, and the mill characteristics [10,11].

It was reported that energy consumption for size reduction of herbaceous feedstocks (corn stover and switchgrass) is 11.0 and 27.6 kWh/metric ton, respectively, while for woody feedstocks (pine and poplar chips) is 85.4 and 118.5 kWh/metric ton, respectively [7].

Tumuluru and Heikkila [12] found in their research that the moisture content of corn stover and grinder speed had a significant influence on the specific energy consumption. They reported that the lower energy consumption value (93 kWh/ton) was observed at a lower moisture content of 10% (wet basis) and a lower grinder speed of 20 Hz. Also, Xia [13] investigated the influence of raw material moisture content on crushing energy consumption. It was found that energy consumption showed an increasing tendency with material moisture content, thus, being necessary to do dry preprocessing before biomass crushing.

Gu et al. [14] studied the influence of planetary ball milling on pre-milled wood fiber (Douglas-fir forest residuals) to improve efficiency of energy consumption. The authors reported that energy consumption for a ball milling ranged from 0.50 kWh/kg (for 7 min milling time) to 2.15 kWh/kg (for 30 min milling time) at 270 rpm.

Hammer milling is widely used for biomass comminution due to its high size reduction ratio and easy adjustment of the particle size range [14]. Usually, the grinding energy for a hammer mill varies between 5 and 60 kWh/ton [12].

Naimi et al. [15] studied the size reduction process using mathematical modelling of particle size distribution of forestry and agricultural biomass. They developed a model that was validated by experimental data, revealing the relation between the grinding rate and the particle size distribution. The main conclusion of the results expressed that for particle size distribution, the normal distribution can be used to predict the first steps of the grinding

process. To predict the other steps of the grinding process it is necessary to examine other distribution types and later define the one that fits better for the analyzed process.

Naimi et al. [16] evaluated the applicability of three industrial-size reduction equations (Kick, Rittinger, and Bond) to the grinding of lignocellulosic biomass in order to estimate the needed energy input to obtain a specific reduction in size. The authors reported that the Rittinger Equation had the best fit to the experimental data and can be used to predict the energy consumption of grinding.

Williams et al. [17] conducted experiments regarding the impact of fresh and dry comminution in a knife mill of biomass pellets on energy consumption, particle size, and shape. The authors concluded that milling energy consumption was significantly lower (19.3–32.5 kW h t$^{-1}$ [fresh] and 17.8–23.2 kW h t$^{-1}$ [dry]) than that reported in literature for non-densified biomass in similar knife mills. They also found that dry grinding reduced milling energy by 38% for mixed wood pellets.

In other research article, Williams et al. [18] studied the impact of four different mills (planetary ball mill, Bond ball mill, knife mill, and ring-roller mill) on the milling behaviour of densified biomass pellets. Their results showed that for optimal milling performance, biomass pellets should be composed of particles which meet the Stokes requirements of the mill classifier.

Shastri et al. [19] studied the optimal level of size reduction and densification for *Miscanthus Giganteus* and switchgrass using the BioFeed, a system-level optimization model. The obtained results showed that 4–6 mm was the optimal particle size for the tested feedstocks. Furthermore, the authors reported that the results of size reduction experiments for both *Miscanthus Giganteus* and switchgrass showed that bulk density and specific energy decreased according to a power law with increasing particle size.

Miao et al. [20] evaluated the comminution energy efficiencies of a commercial-scale hammer mill, a bench-scale Retsch SM2000 knife mill and a Retsch SK100 hammer mill for Miscanthus (*Miscanthus Giganteus*), switchgrass (*Panicum virgatum*), willow (*Salix babylonica*), and energy cane (*Saccharum* spp.) size reduction. The authors reported that biomass moisture significantly influenced comminution energy consumption, especially for finer size reduction. They found that the Retsch SK100 hammer mill was more energy efficient than the SM2000 knife mill due to the higher motor speed and axial feeding mechanism.

Also, Su Dongping and Yu Manlu [21] analyzed the particle size distribution of the grinded material using a hammer mill and a knife mill. Their purpose was to evaluate it by using statistical distribution such as Rosin–Rammler (RR) distribution, normal distribution, and lognormal distribution and then to measure it to the grinding performance of the equipment used. After the analysis, they observed that the distribution that closely matches the results is Rosin Rammler with values of the correlation coefficient R$^2$ between 0.998 and 1. Also, another conclusion revealed that the grinding performance analyzed was better for the knife mill compared to the hammer mill although the experiments were conducted in the same conditions.

The size reduction process was also the subject of another article, where Oyedeji O. et al. [22] presented an evaluation of the biomass size reduction process, trying to insights for interested scientists, engineers, researchers, entrepreneurs for future development and investigation to improve the biomass utilization industry through technologies. Thus, they concluded the importance of studying biomass size reduction in order to produce high-quality grinded particles.

Miranda et al. [23] analyzed the main characteristics of a wide range of pellets made from woody biomass, herbaceous biomass and fruit biomass. The authors evaluated the most representative characteristics (moisture, bulk density, durability, ash content, chemical composition and high heating value) and compared them to specific standard. They reported that their results showed significant differences among the analyzed pellets, exceeding the limits established by the standard. For example, the different results obtained for the pine wastes showed the importance of pretreatments in order to make the most of biomass pellets.

In the same way, Zawislak et al. [24] evaluated the feasibility of using lignocellulosic biomass (chamomile waste, birch sawdust, pea waste, and soybean waste) for pellets production. Their results showed that the biomass composition has a significant effect on the calorific value of pellets and determines the energy consumption of the pellet production process.

In other study, Albashabsheh and Stamm [25] made a literature review for modeling and optimization studies of lignocellulosic biomass supply chains with densification processes. The authors found that baling is the most studied densification technique, while optimization modeling is the most common analysis method.

Another literature review regarding the lignocellulosic biomass pretreatment techniques was realized by the Taylor at al. [26]. The authors concluded that starting with the moisture and particle size reduction of lignocellulosic biomass, the feedstock can undergo a method of physical, physico–chemical or chemical treatment to fully optimize them as a solid fuel. The paper presents the hammer mill working process optimization problem destined for milling energetic biomass (*Miscanthus Giganteus*, and *Salix Viminalis*) taking into consideration the functional and constructive parameters of the hammer mill in order to reduce the specific energy consumption. The analysis for the optimization process was done considering the working parameters, which had to be clarified. The modelling process for optimization was conducted based on objective functions which are of three types: energetic, economic, and production quality. The analysis involved describing the quality of the hammer mill working process.

## 2. Materials and Methods

### 2.1. Hammer Mill and Material Properties

To collect date regarding the energy consumption of a hammer mill, an experimental research was done. The experiment was conducted by using a MC 22 hammer mill (Figure 1) and as feedstock *Miscanthus Giganteus* and *Salix Viminalis* stalks. The hammer mill parameters involve a productivity on grinded material of 0.33–0.50 kg·s$^{-1}$, respectively 0.22–0.42 kg·s$^{-1}$ on grinding packaged hay. The engine of the hammer mill is an electrical engine of 22 kW, using a sieve with $\varnothing$4 mm orifice size (commercial mill variant). Also, the positioning diameter of hammers on the rotor is 220 mm and the hammer mill rotors length is of 500 mm.

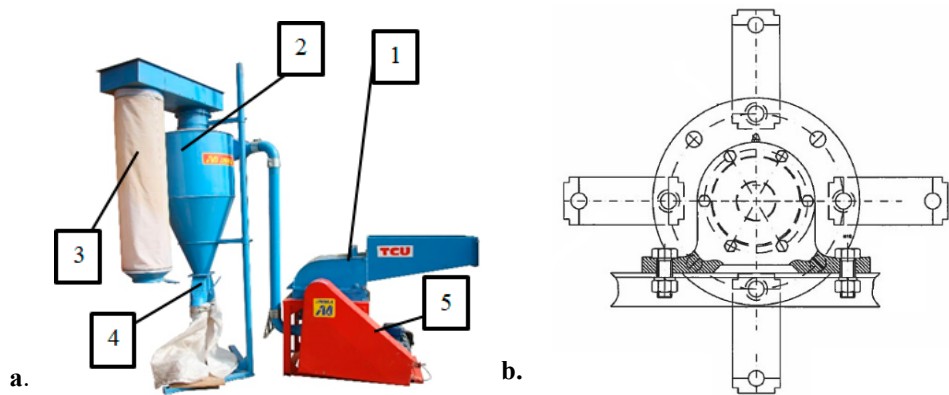

**Figure 1.** (**a**) Hammer mill MC 22 (hammer mill; 2. exhauster; 3. cyclone with support and dust collector bag; 4. grinding material evacuation vent; 5. electrical engine; 6. grinded material.); (**b**) hammer mill rotor [27,28].

The MC 22 hammer mill was equipped with 24 hammers that were set up in a parallel distribution. The hammers used for testing had a length of 153 mm and a width of 60 mm, and they had different edges as it can be seen in Figure 2.

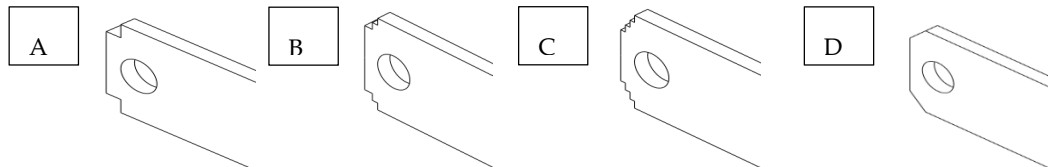

**Figure 2.** The four types of hammers used for testing ((**A**)—one edge step hammer; (**B**)—two edge steps hammer; (**C**)—three edge steps hammer; (**D**)—triangle edge hammer) [27,29].

The dimension of the stalks measured after harvesting process were about 125 mm. The moisture content measured for the samples during the experimental testing (approximately 3 and a half months after harvesting) revealed that the value for *Miscanthus Giganteus* samples were between 8.11–11.31% and a value between 8.89–11.99% was revealed for *Salix Viminalis* [28]. Also, it must be said that for grinding energy consumption we had to consider the following: the material feeding flow values, the dimensions of mill sieve orifices, the type of hammer used for the experiment (four types: one, two and three step edge hammer, triangle edge hammer), and the rotor frequencies applied for testing (50 Hz, 47.5 Hz, 45 Hz, 42.5 Hz, 40 Hz). In general, the peripheral speeds of hammers necessary for grinding lignocellulosic biomass are between 100–115 m/s, values that were considered during experimental research and in this present paper [30–32]. Material flow into the hammer mill differed from one experiment to another, even though the same sieve type was used for multiple experiments. The energy consumption for the grinding process was calculated taking account of current voltage and intensity (triphasic). These were determined using specific measuring equipment (Agilent U1210 Series Handheld Clamp Meters United States), and a data acquisition system for each phase of the voltage networks.

Material granulation was obtained using a classifier RETCH AS 200 basic from Germany with an amplitude of 2.5 mm and during a time of 3 minutes for each sample of biomass grinded. Considering this process in Tables 1 and 2 the material granulation is presented for both *Miscanthus Giganteus* and *Salix Viminalix*.

**Table 1.** Results regarding *Miscanthus Giganteus* grinded material granulation for one step hammer.

| | | *Miscanthus Giganteus* | | | | |
|:---:|:---:|:---:|:---:|:---:|:---:|:---:|
| **Sieve Hole Diameter, (mm)** | **Revolution Speed (Hz)** | **Granulation (g)** | | | | |
| | | **<10 mm** | **10–15 mm** | **15–20 mm** | **20–25 mm** | **>25 mm** |
| 25 | 50 | 1.5177 | 0.6067 | 0.6086 | 0.9054 | 1.1936 |
| 25 | 47.5 | 1.9167 | 1.0990 | 0.6942 | 0.4280 | 0.7476 |
| 25 | 45 | 0.9336 | 0.8517 | 0.9099 | 0.7989 | 1.4655 |
| 25 | 42.5 | 1.0000 | 0.8050 | 0.4780 | 1.1186 | 1.5353 |
| 25 | 40 | 0.6540 | 0.7861 | 0.5085 | 0.8355 | 2.1467 |
| 16 | 50 | 0.3098 | 0.7049 | 0.6494 | 2.3064 | 0 |
| 16 | 47.5 | 0.5172 | 0.7568 | 0.6795 | 2.0316 | 0 |
| 16 | 45 | 0.4454 | 0.7230 | 0.6838 | 2.0706 | 0 |
| 16 | 42.5 | 0.5484 | 0.4900 | 0.6707 | 2.2075 | 0 |
| 16 | 40 | 0.5691 | 0.4619 | 0.7572 | 2.1748 | 0 |
| 10 | 50 | 0.9181 | 0.9999 | 1.0592 | 0.0000 | 0.0000 |
| 10 | 47.5 | 0.8336 | 1.0580 | 1.0848 | 0.0000 | 0.0000 |
| 10 | 45 | 0.9252 | 0.8905 | 1.1632 | 0.0000 | 0.0000 |
| 10 | 42.5 | 0.5295 | 0.6967 | 1.7522 | 0.0000 | 0.0000 |
| 10 | 40 | 0.8326 | 1.0004 | 1.1456 | 0.0000 | 0.0000 |

**Table 2.** Results regarding *Salix Viminalis* grinded material granulation for one step hammer.

| Sieve Hole Diameter, (mm) | Revolution Speed (Hz) | Granulation (g) | | | |
|---|---|---|---|---|---|
| | | <5 mm | 5–10 mm | 1016 mm | >16 mm |
| 16 | 50 | 2.9980 | 3.1526 | 3.3808 | 1.2987 |
| 16 | 47.5 | 2.3693 | 3.4656 | 3.3416 | 0.7442 |
| 16 | 45 | 0.9982 | 3.3333 | 4.9144 | 0.6499 |
| 16 | 42.5 | 1.2357 | 3.2332 | 4.7799 | 0.6438 |
| 16 | 40 | 1.2812 | 2.7812 | 5.1520 | 0.6793 |
| 10 | 50 | 0.6112 | 2.2733 | 1.8222 | 1.2585 |
| 10 | 47.5 | 1.2909 | 1.8832 | 1.4341 | 1.3468 |
| 10 | 45 | 1.6824 | 1.8411 | 1.4851 | 0.9510 |
| 10 | 42.5 | 1.2995 | 1.6210 | 1.5000 | 1.5383 |
| 10 | 40 | 0.9550 | 1.1583 | 1.6525 | 2.1920 |
| 7 | 50 | 0.7376 | 2.5000 | 1.7284 | 0 |
| 7 | 47.5 | 1.1920 | 2.5630 | 1.2090 | 0 |
| 7 | 45 | 1.1375 | 2.4414 | 1.3910 | 0 |
| 7 | 42.5 | 1.0055 | 2.6430 | 1.3165 | 0 |
| 7 | 40 | 1.8980 | 2.0475 | 1.0912 | 0 |

To better illustrate the granulation, Figure 3 presents the material granulation (grinded material) from the experiments for *Miscanthus Giganteus* as well for *Salix Viminalis*, for a revolution speed of 50 Hz and the three-step edge hammer.

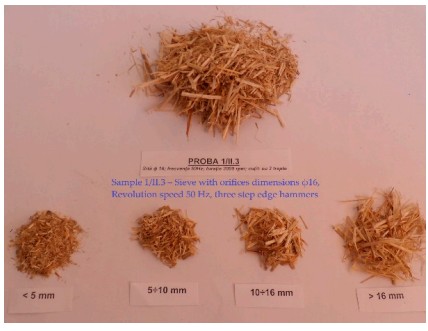
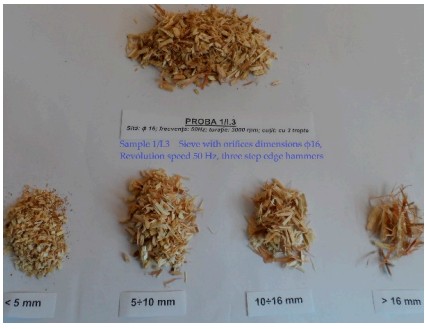

a.                                                                                          b.

**Figure 3.** Material subjected to grinding process using a three-step edge hammer and a revolution speed of 50 Hz: (**a**) *Miscanthus Giganteus*; (**b**) *Salix Viminalis*.

To better describe the process and the relations between the output and the input of the hammer mill process an analysis regarding hammer mill working process optimization was done considering the working parameters of the hammer mill. This type of analysis was selected because the working process of a hammer mill has an undifferentiated working process character. To illustrate de findings in Tables 1 and 2 the granulation of the grinded material is presented for one step hammer.

The sieve diameter, revolution speed, and feeding flow were some of the parameters considered for the statistical optimization of the hammer mill working process. These data were used by the authors for different other articles pursuing other objectives [29,33–35]. The table below (Tables 3 and 4) presents the consumed power and the measured grinded particle size, considering different sieve hole diameters, revolution speeds, and feeding flow, four types of hammers, and two biomass types used for the grinding process for *Miscanthus Giganteus* and *Salis Viminalis*.

**Table 3.** Results obtained during experimental tests for *Miscanthus Giganteus* [34].

| | | | | | *Miscanthus Giganteus* | | | | |
|---|---|---|---|---|---|---|---|---|---|
| Sieve Hole Diameter, (mm) | Revolution Speed (Hz) | Material Flow (kg·s$^{-1}$) | Consumed Power (kW) | Grinded Particle Diameter (mm) | Sieve Hole Diameter, (mm) | Revolution Speed (Hz) | Material Flow (kg·s$^{-1}$) | Consumed Power (kW) | Grinded Particle Diameter (mm) |
| Hammer with one-edge corners | | | | | Hammer with three-edge corners | | | | |
| 25 | 50 | 0.144 | 13.31 | 17.65 | 25 | 50 | 0.294 | 17.54 | 23.08 |
| 25 | 47.5 | 0.185 | 13.17 | 14.29 | 25 | 47.5 | 0.357 | 14.69 | 21.00 |
| 25 | 45 | 0.214 | 11.69 | 20.17 | 25 | 45 | 0.312 | 9.54 | 22.01 |
| 25 | 42.5 | 0.149 | 8.02 | 20.54 | 25 | 42.5 | 0.277 | 9.74 | 23.64 |
| 25 | 40 | 0.128 | 7.47 | 23.29 | 25 | 40 | 0.294 | 11.09 | 22.02 |
| 16 | 50 | 0.224 | 9.65 | 18.04 | 16 | 50 | 0.25 | 16.08 | 14.22 |
| 16 | 47.5 | 0.227 | 9.28 | 16.65 | 16 | 47.5 | 0.151 | 14.19 | 16.15 |
| 16 | 45 | 0.135 | 6.55 | 16.80 | 16 | 45 | 0.208 | 12.39 | 18.07 |
| 16 | 42.5 | 0.121 | 5.21 | 17.24 | 16 | 42.5 | 0.217 | 12.18 | 16.62 |
| 16 | 40 | 0.128 | 7.13 | 17.28 | 16 | 40 | 0.166 | 11.30 | 13.07 |
| 10 | 50 | 0.217 | 12.96 | 8.56 | 10 | 50 | 0.161 | 16.59 | 9.53 |
| 10 | 47.5 | 0.192 | 9.96 | 8.76 | 10 | 47.5 | 0.147 | 14.65 | 9.64 |
| 10 | 45 | 0.166 | 7.96 | 8.81 | 10 | 45 | 0.142 | 13.75 | 9.50 |
| 10 | 42.5 | 0.166 | 10.74 | 10.94 | 10 | 42.5 | 0.156 | 10.87 | 10.07 |
| 10 | 40 | 0.116 | 7.64 | 8.92 | 10 | 40 | 0.125 | 12.21 | 10.14 |
| Hammer with two–edge corners | | | | | Hammer with oblique corners | | | | |
| 25 | 50 | 0.25 | 15.81 | 17.92 | 25 | 50 | 0.263 | 13.62 | 22.33 |
| 25 | 47.5 | 0.25 | 13.03 | 18.81 | 25 | 47.5 | 0.166 | 12.17 | 23.37 |
| 25 | 45 | 0.208 | 11.54 | 16.31 | 25 | 45 | 0.161 | 10.64 | 24.17 |
| 25 | 42.5 | 0.147 | 11.54 | 21.03 | 25 | 42.5 | 0.166 | 14.48 | 24.81 |
| 25 | 40 | 0.2 | 11.30 | 21.26 | 25 | 40 | 0.178 | 13.02 | 23.86 |
| 16 | 50 | 0.172 | 19.70 | 15.48 | 16 | 50 | 0.312 | 13.98 | 14.00 |
| 16 | 47.5 | 0.1928 | 19.64 | 14.85 | 16 | 47.5 | 0.238 | 12.47 | 16.72 |
| 16 | 45 | 0.166 | 12.81 | 15.87 | 16 | 45 | 0.294 | 11.22 | 20.00 |
| 16 | 42.5 | 0.142 | 11.55 | 15.62 | 16 | 42.5 | 0.208 | 14.37 | 18.26 |
| 16 | 40 | 0.166 | 8.07 | 16.68 | 16 | 40 | 0.125 | 8.81 | 17.45 |
| 10 | 50 | 0.185 | 17.59 | 8.70 | 10 | 50 | 0.192 | 13.84 | 11.04 |
| 10 | 47.5 | 0.142 | 10.34 | 8.95 | 10 | 47.5 | 0.151 | 13.87 | 11.17 |
| 10 | 45 | 0.192 | 16.96 | 9.42 | 10 | 45 | 0.138 | 15.86 | 10.71 |
| 10 | 42.5 | 0.166 | 11.18 | 9.45 | 10 | 42.5 | 0.111 | 11.15 | 10.85 |
| 10 | 40 | 0.116 | 11.22 | 10.63 | 10 | 40 | 0.108 | 9.61 | 11.10 |

Table 4. Results obtained during experimental tests for *Salix Viminalis*.

| Sieve Hole Diameter, (mm) | Revolution Speed (Hz) | Material Flow (kg·s$^{-1}$) | Consumed Power (kW) | Grinded Particle Diameter (mm) | Sieve Hole Diameter, (mm) | Revolution Speed (Hz) | Material Flow (kg·s$^{-1}$) | Consumed Power (kW) | Grinded Particle Diameter (mm) |
|---|---|---|---|---|---|---|---|---|---|
| | | Hammer with one–edge corners | | | | | Hammer with three–edge corners | | |
| 16 | 50 | 0.385 | 11.73 | 10.40 | 16 | 50 | 0.4 | 13.10 | 9.24 |
| 16 | 47.5 | 0.417 | 11.82 | 9.40 | 16 | 47.5 | 0.333 | 11.19 | 10.27 |
| 16 | 45 | 0.385 | 10.32 | 10.76 | 16 | 45 | 0.333 | 10.54 | 9.85 |
| 16 | 42.5 | 0.313 | 12.77 | 10.56 | 16 | 42.5 | 0.286 | 8.29 | 10.95 |
| 16 | 40 | 0.278 | 9.09 | 10.80 | 16 | 40 | 0.308 | 6.43 | 11.51 |
| 10 | 50 | 0.333 | 14.27 | 8.02 | 10 | 50 | 0.286 | 13.59 | 7.41 |
| 10 | 47.5 | 0.385 | 15.48 | 7.56 | 10 | 47.5 | 0.267 | 12.62 | 8.05 |
| 10 | 45 | 0.417 | 14.87 | 6.73 | 10 | 45 | 0.286 | 7.98 | 8.29 |
| 10 | 42.5 | 0.313 | 14.23 | 7.89 | 10 | 42.5 | 0.286 | 8.47 | 7.81 |
| 10 | 40 | 0.2 | 9.91 | 9.20 | 10 | 40 | 0.267 | 10.81 | 8.21 |
| 7 | 50 | 0.417 | 17.07 | 6.50 | 7 | 50 | 0.267 | 15.40 | 5.30 |
| 7 | 47.5 | 0.238 | 13.72 | 5.71 | 7 | 47.5 | 0.267 | 13.64 | 6.10 |
| 7 | 45 | 0.417 | 12.81 | 5.92 | 7 | 45 | 0.444 | 11.64 | 5.83 |
| 7 | 42.5 | 0.295 | 12.44 | 5.94 | 7 | 42.5 | 0.286 | 11.11 | 6.07 |
| 7 | 40 | 0.357 | 9.70 | 5.19 | 7 | 40 | 0.191 | 7.64 | 6.22 |
| | | Hammer with two–edge corners | | | | | Hammer with oblique corners | | |
| 16 | 50 | 0.357 | 15.73 | 9.51 | 16 | 50 | 0.231 | 13.72 | 11.65 |
| 16 | 47.5 | 0.417 | 13.80 | 9.75 | 16 | 47.5 | 0.25 | 10.24 | 11.56 |
| 16 | 45 | 0.357 | 10.57 | 9.82 | 16 | 45 | 0.25 | 8.75 | 11.94 |
| 16 | 42.5 | 0.333 | 8.53 | 11.62 | 16 | 42.5 | 0.2 | 7.28 | 11.35 |
| 16 | 40 | 0.263 | 7.55 | 11.16 | 16 | 40 | 0.176 | 6.03 | 9.77 |
| 10 | 50 | 0.556 | 18.96 | 7.36 | 10 | 50 | 0.333 | 12.28 | 7.57 |
| 10 | 47.5 | 0.500 | 16.71 | 7.56 | 10 | 47.5 | 0.375 | 11.46 | 8.52 |
| 10 | 45 | 0.455 | 13.26 | 7.65 | 10 | 45 | 0.333 | 8.98 | 7.50 |
| 10 | 42.5 | 0.357 | 12.84 | 7.80 | 10 | 42.5 | 0.333 | 7.97 | 8.87 |
| 10 | 40 | 0.357 | 10.13 | 7.59 | 10 | 40 | 0.214 | 7.61 | 9.31 |
| 7 | 50 | 0.500 | 16.72 | 5.98 | 7 | 50 | 0.333 | 11.31 | 6.07 |
| 7 | 47.5 | 0.500 | 17.54 | 5.40 | 7 | 47.5 | 0.3 | 8.28 | 6.59 |
| 7 | 45 | 0.500 | 14.82 | 5.55 | 7 | 45 | 0.273 | 9.51 | 6.43 |
| 7 | 42.5 | 0.333 | 15.91 | 5.84 | 7 | 42.5 | 0.231 | 9.74 | 6.42 |
| 7 | 40 | 0.25 | 11.26 | 6.10 | 7 | 40 | 0.2 | 7.24 | 5.68 |

## 2.2. Correlations between the Parameters Which Describe the Hammer Mill Working Process

In order to realize the hammer mill working process for biomass grinding, firstly, some aspects must be clarified. Thus, when analyzing the experiments, 12 parameters defining the hammer mill working process were defined, out of which ($q$, $m$, $t$, $P$, $U$, $I$, $E$, $\varepsilon$) are linked, and 4 are free ($d_s$, $v$, $u$, $g_r$). According to the experimental research, the parameters were classified as follows:

- input parameters—material connected: material mass before grinding $m$ (kg), time necessary for grinding $t$ *(s)*, feeding flow $q$ (kg·s$^{-1}$), moisture content $u$ (%);
- electrical energy feeding flow parameters: Power $P$, tension $U$, electrical current intensity $I$, consumed electrical energy $E$, specific energy $\varepsilon$ (the energy reported for the grinded material mass);
- command and control parameters: sieve orifice diameter $d_s$, rotation frequency $v$;
- output parameters—connected to the quality of the grinded material: graininess $g_r$ (mm).

In order to describe internal system relations and the relations between the output, control, and input dimensions, there are no physical applicable relations. This situation is because of the random character of the grinding process inside the hammer mill, due to the variety of parameters that influences the process. The feeding material, which in this case are vegetal stems (with a certain resistance, influenced by humidity), is introduced in the working chamber, and subjected to a process of random movement in which biomass undergoes shocks provoked by hammers leading to shearing (in general), or ruptured in conditions that are lesser known or describable. The process continues until the entire material load is grinded to the required dimension and passes through the sieve. In these conditions, the relations which mathematically describe the connections between system parameters are researched through statistical modeling. More precisely, given our specific case, this refers to obtaining the best statistical regressions, for the main process quality parameters. After this, correlations are used for seeking and identifying eventual optimal points inside the process parametric space.

## 2.3. Considerations Regarding Hammer Mill Working Process Optimization

Taking the classification of hammer mill working process parameters into consideration, we can assume the optimization parameters and the most likely objective functions. Restrictions, if there is a case for them, must be outlined beforehand, but can also be added afterwards, and then the optimization algorithm can be finely tuned.

As optimization parameters, meaning arguments for the objective functions, it is mandatory to consider the command or control parameters of the process, meaning the frequency $v$ (respectively the hammer mill rotor speed), and the diameter of the sieve orifices in use, $d_s$.

The rotor speed is the command that gives the speed of the system. The feeding current tension $U$ and intensity $I$, are dimensions which are measured. Using the measured values for the current tension and intensity, power and energy can be calculated, respectively the energy of the processed mass unit $\varepsilon$ (J/kg). The diameter of the sieve orifices being used $d_s$ was chosen, mainly for calibrating the maximum length of the grinded material.

A parameter that can also be taken into consideration, as an argument of the objective functions, is the material humidity, $u$ (%). This can be considered an input parameter linked with state of the feeding material, and can also be considered as a control parameter, if, concluding the existence of an optimum processing humidity figure can lead to a desired drying process for the feeding material before introducing it in the hammer mill.

A double standard is given to the input parameters: mass $m$ and processing time $t$. Firstly, they are arguments for the energy type objective function (as example for $\varepsilon$), and the flow $q$, calculated using the two parameters, which can become an argument for the complex objective function (specific energy for the working capacity unit). On the other hand, the flow, as calculated in this model, is a measure of the working capacity (disregarding the discharge times, maintenance, etc.), thus meaning an objective function.

The main parameters for the hammer mill working regime, in this research, are confirmed by a large part of the specialty literature [36–43].

Objective functions taken into consideration in this modeling stage (exceptions to objective functions like endurance, viability, etc.) are of three types: energetic, economical, and production quality.

Objective functions of the energetic type are:

- energy $E$ calculated from parameters $U$ and $I$, and having as arguments parameters $v$, $d_s$, $u$, eventually $m$ and $t$, or synthetic $q$;
- energy $\varepsilon$ calculated from parameters $U$ and $I$, and having as arguments parameters $v$, $t$, $u$, eventually $m$ and $t$, or synthetic $q$;
- unit working capacity specific energy $w$, calculated with the relation:

$$w = \frac{E}{q} = \frac{E}{m}\, t = \varepsilon t \tag{1}$$

From the category of the economical type objective functions, only the working capacity $q$ was chosen, in this case the feeding flow in the working process. We can use as arguments the speed (hammer mill rotor speed $v$), the diameter of the sieve orifices and the feeding material humidity. There are other economical type objective functions, linked with energy and quality of the feeding material, but they require extensive knowledge regarding feeding material, processing, and energy costs.

The objective function that must maximize the production quality is described by the distribution of grinded material on dimensions and has as arguments the mill rotor frequency (speed), diameter of the sieve orifices being used, material feeding flow, and the material humidity.

The objective function proposed for study is considered in most specialty research papers, not only for the hammer mills, but also in the problem of optimization and evaluation of mill functionality in general [44–50], as mentioned before.

### 2.4. Objective Function Which Describes the Hammer Mill Working Process Quality Destined for Grinding

The realized experiments for grinding vegetal material using hammer mills equipped with different types of hammers, had as an objective the sorting of the resulted material. In general, there are no standardized demands for the component segment lengths for the obtained grind using hammer mills. According to the demand of a beneficiary and the purpose for which the grinded material will be used, a demand for a certain dimension or, in general, an interval of material fragment lengths can be imposed. This means that from a qualitative point of view, obtaining a required length for the grinded material segments, close to the requested dimensions by the beneficiary, is set as a target, and of course, a certain minimum percentage of the grinded material must be according to that value (70–95%). Also, from experimental analysis, there are some material losses, which, in case they are significant, must be reduced. Reaching a certain grinded material dimension, depends on the working parameters being considered (control parameters: mill speed—rotor frequency, feeding flow, the orifices dimensions of the sieve used for evacuating the grinded material).

If the mill is working with discontinuous feeds, in a variable time, only until the entire material quantity passes through the fixed sieve, can we precisely appreciate the flow, and fix a maximum dimension for the grind, requested by a beneficiary. However, if using the sieve with the maximum orifice size, the distribution of gridded segment length is practically impossible to control for a dimension smaller than the sieve orifices being used.

In the authors' opinion, data regarding grinded material granulation, normalized through a division on the processed mass (and eventually a multiplication by 100 for a percentage representation) represents an approximation of a probability density for obtaining a granulation between certain intervals.

If the values are cumulated (practically integrating the probability density), we can obtain the probability to obtain a grind in the lower segments of a fixed value.

A beneficiary can ask for a lower grind value than a given dimension. In this case, if we can obtain by interpolation a function which can approximate the probability density, the operator can choose a working regime, meaning a sieve speed (with certain orifice sizes), and a feeding flow, which can permit a smaller dimension, or at most equal to the given figure by the beneficiary.

The creation of a direct function on the existent normalized data was researched, through the interpolation of the probability density. A second degree polynomial interpolation with four variables was realized, where $P = P(x, d_s, v, q)$, where $x$ is the maximum dimension of the grinded segments:

$$P(x, d_s, v, q) = P_0 + a_1 x + a_2 x^2 + b_1 d_s + b_2 d_s^2 + c_1 v + c_2 v^2 + d_1 q + d_2 q^2 + a_3 x d_s + a_4 x v + a_5 x q + b_3 d_s + b_4 d_s q + c_3 v q \quad (2)$$

This interpolation did not give fully satisfactory results (correlation between original and obtained data through interpolation was 0.671, and the maximum error was of approx. 65%). For these reasons, the same type of interpolation was realized for the probability of obtaining a grind with smaller dimensions than the x dimension, proposed by the beneficiary. We obtained the function (3) and correlating it with the experimental data lead to a figure of 0.021, a maximum error of 29.59%, considering acceptable values.

Thus, the function that approximates through interpolation (using the method of the smallest squares) the probability that the grinded particles have the maximum dimension inferior to a given figure, is of the following type:

$$P(x, d_s, v, q) = 545.009 + 3945.389x - 0.642x^2 - 852.676d_s \\ -244158.187d_s^2 - 28.399v + 0.24v^2 + 949.638q \\ -7802.92q^2 + 155488.247xd_s - 143.957xv + 17010.48xq \\ +181.595d_sv - 0.165d_sq + 34.85vq \quad (3)$$

## 3. Results and Discussions

### 3.1. Applying the Objective Function for Grinded Material Quality Control

In Figures 4 and 5, the experimental data regarding the percentual particle size (grind) distribution obtained during grinding miscanthus are presented. There were no experiments done using the sieve with orifice dimensions of φ7 mm.

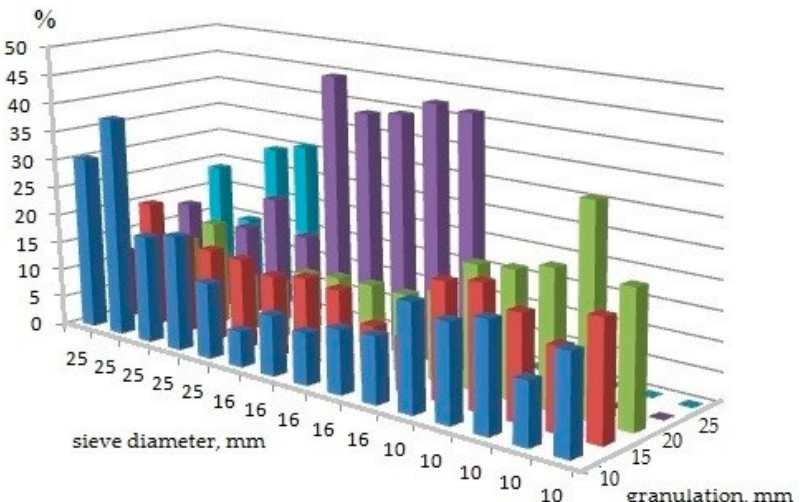

**Figure 4.** Graphical representation of experimental data regarding grinded *Miscanthus Giganteus* granulation (for each frequency in Tables 1 and 2).

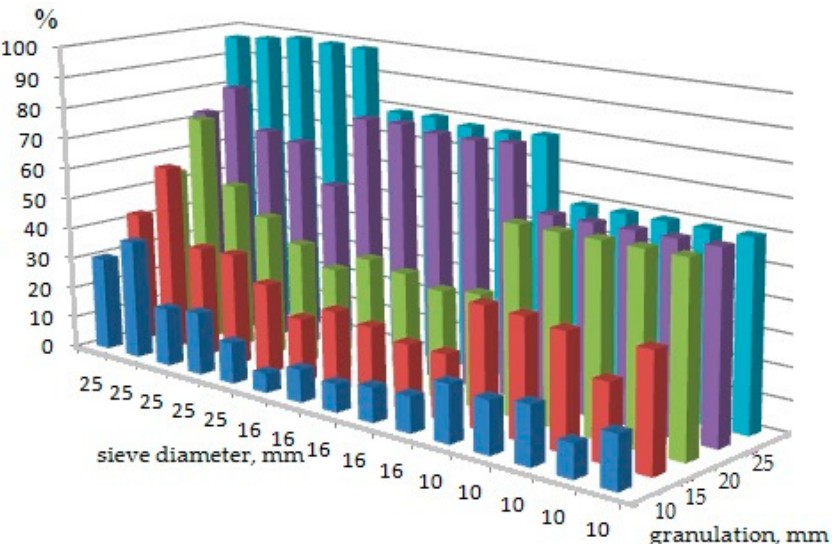

**Figure 5.** Graphical representation of cumulated experimental data regarding grinded *Miscanthus Giganteus* granulation (for each frequency in Tables 1 and 2).

Granulation columns from the experimental data tables (Tables 1 and 2) were added, and we obtained the experimental distribution probability so that the grinded particles will have the maximum dimension inferior to a given dimension. We can observe that there are losses (differences from the percentage of 100%) (Figure 6), that, by themselves, can form the subject of an objective function needing to be minimized. Also, in Figure 6b the average values and the standard deviation are shown.

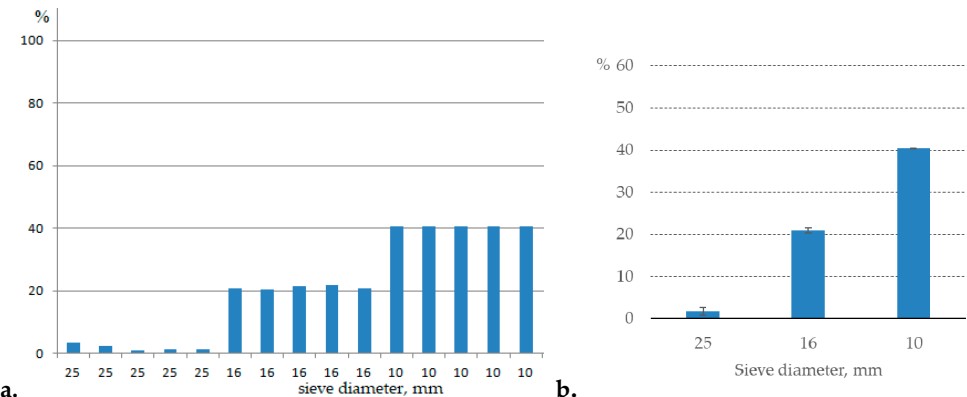

**Figure 6.** Variation of the losses (differences from 100%) represented using *Miscanthus Giganteus* experimental data (speed revolution 50 Hz, particle size > 25 mm), for the five fractions obtained for particle size reduction analysis (according to Table 1) (absolute values—(**a**); average values—(**b**)).

Taking advantage of the function (3), for establishing an optimal working regime, means using it together with an energetic type objective function, example Equation (4).

$$
\begin{aligned}
E = E(v, d_s, q) = \quad & E_o + e_1 d_s + e_2 v + e_3 q + e_{12} v d_s \\
& + e_{23} v q + e_{13} q\, d_s + e_{11} d_s^2 + e_{22} v^2 \\
& + e_{33} q^2
\end{aligned}
\tag{4}
$$

Thus, let us assume that the beneficiary requires a high percentage of grinded material, with the average dimension smaller than an x value. In this case, the following optimal problem must be resolved:

$$P(x_o, d_s, v, q) \rightarrow \ max, E(d_s, v, q) \rightarrow min \tag{5}$$

The triplets $(d_s, v, q)$, if present, must be found in order to minimize the energetic objective function, and at the same time, maximize the probability of obtaining the required granulation.

### 3.2. Synthesis of Applying Equation Model (2) for the Four Types of Hammers of the Hammer Mill

In this section, the results of applying calculus model (2) are presented, for quality characteristics of all four variants of hammers under examination, using as grinding material *Miscanthus Giganteus* stems mechanical harvested.

From the analysis of the Table 3 data, we can conclude, that regarding the process influence, as absolute size, the polynomial equation coefficients indicate an intense dependency of Function (2) to the figures $d_s^2, d_s, xd_s, q$.

As an order of size, coefficient distribution is similar for the four types of hammers: signals (+ or −) differ for some coefficients, but not for the dominant ones.

Coherency with the case in which *Miscanthus Giganteus* stems are grinded is evident when it comes to interpolation polynomic structure. The coefficients and quality solution estimation performance for Equation (2) for both *Miscanthus Giganteus* and *Salix Viminalis* are presented in Tables 5 and 6.

**Table 5.** Coefficients and quality solution estimation performance for Equation (2) for the four types of hammers being used at grinding *Miscanthus Giganteus* stems.

| Coeff. Equation (2) | One Step Edge Hammer | Two Step Edge Hammer | Three Step Edge Hammer | Triangular Edge Hammer |
|---|---|---|---|---|
| $P_0$ | 545.009 | 7.647 | 128.178 | −191.809 |
| $a_1$ | 3945.389 | 1840.053 | −773.96 | 4381.88 |
| $b_1$ | −852.676 | 11,921.83 | 8528.7 | 13,948.771 |
| $c_1$ | −28.399 | −1.153 | −8.827 | 5.781 |
| $d_1$ | 949.638 | −998.537 | 250.614 | −1068.779 |
| $a_3$ | 155,488.247 | 135,065.701 | 260,449.374 | 137,745.435 |
| $a_4$ | −143.957 | −60.465 | 19.463 | −128.117 |
| $a_5$ | 17,010.48 | 7892.295 | −7283.796 | 12,028.786 |
| $b_3$ | 181.595 | −114.671 | 25.341 | −20.23 |
| $b_4$ | −0.165 | −0.228 | −0.065 | 0.043 |
| $c_3$ | 34.85 | 37.241 | −4.106 | 25.526 |
| $a_2$ | −0.642 | −0.803 | 0.316 | 0.314 |
| $b_2$ | −244,158.187 | −233,661.571 | −395,343.692 | −409,821.124 |
| $c_2$ | 0.240 | −0.03 | 0.097 | −0.072 |
| $d_2$ | −7802.921 | −1786.404 | 288.491 | −1029.165 |
| Correlation with experimental data | 0.921 | 0.911 | 0.902 | 0.911 |
| Maximum error, % | 29.621 | 33.895 | 30.185 | 28.99 |

**Table 6.** Coefficients and quality solution estimation performance for Equation (2) for the four types of hammers being used at grinding *Salix Viminalix* stems.

| Coeff. Equation (2) | One Step Edge Hammer | Two Step Edge Hammer | Three Step Edge Hammer | Triangular Edge Hammer |
|---|---|---|---|---|
| $P_0$ | 2386.951 | 315.483 | 339.318 | 548.367 |
| $a_1$ | −18,891.136 | 6213.758 | 3442.265 | −1694.161 |
| $b_1$ | −106,124.527 | −41,580.718 | −39,053.984 | −48,975.424 |
| $c_1$ | −79.633 | −7.279 | −6.539 | −8.178 |
| $d_1$ | 91.056 | 64.421 | −251.411 | −791.751 |
| $a_3$ | 1,081,578.733 | 600,334.1 | 793,974.298 | 1,011,981.53 |
| $a_4$ | 396.421 | −132.5 | −57.749 | 110.325 |
| $a_5$ | −6542.581 | 1295.849 | −726.819 | −6111.094 |
| $b_3$ | 1394.333 | 261.058 | 36.363 | −75.148 |
| $b_4$ | −1.896 | −1.872 | −2.211 | −2.008 |
| $c_3$ | 5.9 | 1.584 | 9.648 | 13.636 |
| $a_2$ | −8.273 | −0.503 | 0.15 | −0.103 |
| $b_2$ | 1,550,328.433 | 1,193,931.883 | 1,445,038.237 | 2,057,669.171 |
| $c_2$ | 0.671 | 0.065 | 0.055 | 0.039 |
| $d_2$ | −452.861 | −150.382 | −229.737 | 541.331 |
| Correlation with experimental data | 0.929 | 0.978 | 0.980 | 0.985 |
| Maximum error, % | 29.487 | 11.386 | 10.539 | 8.651 |

Regarding the influence in this process, as an absolute measure, the polynomial coefficients show a high dependence of the Equation (3) to terms $x$, $d_s$, $v$, $q$.

### 3.3. Grinded Material Quality Study with the Help of Statistical Distribution

In this section, the use of quality objective function is described, deduced from the smallest squares method using experimental data. As it was previously shown, the best form of the quality objective function is a second degree four variable polynomial type.

Function $P$ given in relation (2) represents the percentage of granular material of a smaller size then the $x$ length. Function (2) can be regarded as the probability that in the controlled process at the sieve with $d_s$ diameter, rotor frequency $v$ and feeding flow $q$, fragments with a smaller size than $x$ must be produced. This probability can be maximized to the three process commands $d_s$, $v$, $q$.

Coordinates of the maximum point will be dependent on $x$ parameter, meaning the maximum limit size of the fragments. In other words, the maximum coordinate point $(d_{smax}(x), v_{max}(x), q_{max}(x), P_{max}(x))$, is dependent on $x$.

Noting:

$$\Delta = \begin{vmatrix} 2b_2 & b_3 & b_4 \\ b_3 & 2c_2 & c_3 \\ b_4 & c_3 & 2d_2 \end{vmatrix} \quad \Delta d_s(x) = \begin{vmatrix} -b_1 & -a_3x & b_3 & b_4 \\ -c_1 & -a_4x & 2c_2 & c_3 \\ -d_1 & -a_5x & c_3 & 2d_2 \end{vmatrix}$$

$$\Delta v(x) = \begin{vmatrix} 2b_2 & -b_1 & a_3x & b_4 \\ b_3 & -c_1 & a_4x & c_3 \\ b_4 & -d_1 & a_5x & 2d_2 \end{vmatrix} \quad \Delta q(x) = \begin{vmatrix} 2b_2 & b_3 & -b_1 & -a_3x \\ b_3 & 2c_2 & -c_1 & -a_4x \\ b_4 & c_3 & -d_1 & -a_5x \end{vmatrix} \tag{6}$$

we obtained the following maximum point coordinate expressions (which maximize the probability to obtain grinding fragments with a smaller size than x):

$$d_{smax}(x) = \frac{\Delta d_s(x)}{\Delta}, \quad v_{max}(x) = \frac{\Delta v(x)}{\Delta}, \quad q_{max}(x) = \frac{\Delta q(x)}{\Delta} \tag{7}$$

Distribution of the grinded material (the main result of the grinding process) is characterized, in general, statistically, showing how the hammer mill can reduce the geometrical dimensions of the processed material, sometimes searching a forecast of the distribution regarding the process command parameters. One of the statistical distributions most used in such problems is the Rosin–Rammler distribution, used for example in [51].

Thus, we will consider o Rosin–Rammler distribution, of the form:

$$P(x, d_s, v, q) = 1 - exp\left[-\left(\frac{x}{a(d_s, v, q)}\right)^{b(d_s, v, q)}\right] \tag{8}$$

in which *a* and *b* are functions of considered command parameters:

$$a(d_s, v, q) = a_0 d_s + a_1 v + a_2 q \, b(d_s, v, q) = b_0 d_s + b_1 v + b_2 q \tag{9}$$

Functions *a* and *b* can have the form (9) or any other adequate form (there is an infinity of such possibilities). For using a hammer mill with one step edges, in case the grinded material is *Miscanthus Giganteus*, the following figures of regression coefficients are obtained $a_0 = -0.793$, $a_1 = 0.001$ m·s, $a_2 = -0.132$ m·s/kg, $b_0 = -0.574$ m$^{-1}$, $b_1 = 0.04$ s, $b_2 = 0.005$ s/kg (Equation (8)).

For these values, we represented the cumulative Rosin–Rammler probability curves in Figure 7.

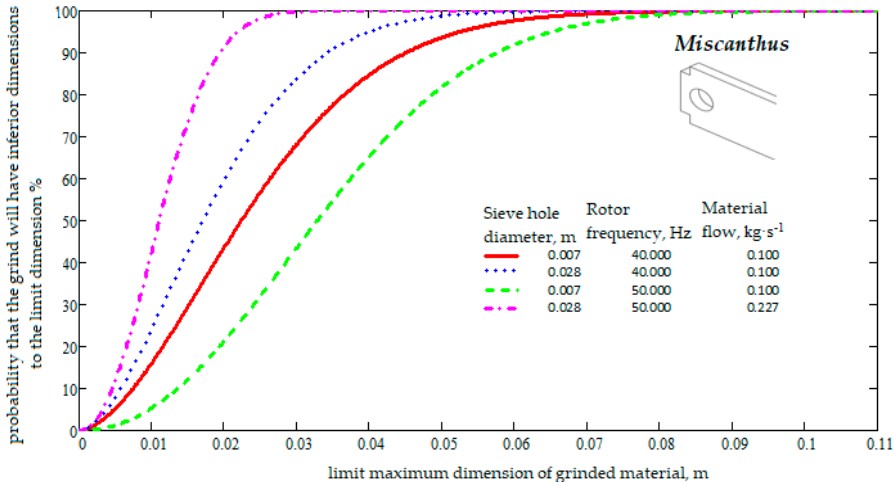

**Figure 7.** Cumulative theoretical probability that the grind will have inferior dimensions to the limit dimension (for four combinations of command parameters: $(x, d_s, v, q)$, drawn based on experimental data obtained during grinding *Miscanthus Giganteus* stems.

The graphical representation in Figure 7 is interpreted as it follows: ground segments of a dimension smaller than 0.11 m are always obtained, meaning a probability 1 or 100%, segments larger than 0.05 m, in the case of using 7 mm orifice size sieves, at a frequency of 50 Hz and feeding flow of 0.1 kg·s$^{-1}$, are found in the grinded material with the probability of 38%. If the 28 mm orifices dimension sieve is used, at frequency of 50 Hz and feeding flow of 0.227 1 kg·s$^{-1}$, the segments with dimensions higher than 5 cm, are with a probability of under 1%.

Synthetic results for the other cases, in which the parameters of Rosin–Rammler distribution (8) and (9) where calculated by minimizing the functional (10), found in Table 7.

$$\Omega(a_o, a_1, a_2, b_0, b_1, b_2) = \sum_{k=0}^{n} [M(x_i, d_{s.i,i}, q_i, a_0, a_1, a_2, b_0, b_1, b_2) - g_i]^2 \qquad (10)$$

**Table 7.** Statistical qualitative model using Rosin–Rammler distribution (rel. 10).

| Hammer Type | $a_0$ | $a_1$ | $a_2$ | $b_0$ | $b_1$ | $b_2$ |
|---|---|---|---|---|---|---|
| *Miscanthus Giganteus* | | | | | | |
| One step edge | −0.314 | 0.001 | −0.152 | 0.349 | 0.043 | 0.005 |
| two step edge | 0.389 | 0.001 | −0.202 | 0.138 | 0.041 | 0.005 |
| three step edge | −0.496 | 0.001 | 0.005 | 0.493 | 0.039 | 0.005 |
| triangular edge | −0.194 | 0.001 | −0.065 | 0.689 | 0.045 | 0.005 |
| *Salix Viminalis* | | | | | | |
| One step edge | 0.273 | 0.000 | −0.011 | 1.000 | 0.055 | 0.007 |
| two step edge | 0.229 | 0.000 | −0.003 | 1.000 | 0.056 | 0.007 |
| three step edge | 0.247 | 0.000 | −0.006 | 1.000 | 0.054 | 0.010 |
| triangular edge | 0.349 | 0.000 | 0.018 | 0.922 | 0.045 | 0.006 |

The distribution of the grinded material sizes inside hammer mills, appears for example in [51].

Unlike our study on the grinded material length distribution, the article refers only on the grinded material distribution, in relation to the regime parameters of the knife mill. The authors of the present paper gave probability functions for obtaining a granulation of the required size, according to a customer's demand, thus, optimizing the quality of the product in the grinding process.

According to [51], the dimension of the sieve orifices and the feeding flow influence the grinded material distribution the most, while the rotor frequency indicates a weaker relation with the same distribution.

The results showed that if the mill is equipped with the 25 mm orifice sieve, we could obtain average particle sizes of 19.1 mm, using one step hammers or two step hammers, 22 mm for three step hammers, respectively 23.8 mm for the triangle edge hammers, using feeding flows between 0.15–0.2 kg/s and hammer rotor speeds between 40–50 Hz. In connection, the specific energy consumption was even 17–18% smaller in the case of using one step hammers, in comparison to the next levels, which were given by two step hammers or with approximately 20% towards the three step hammers, these results being recorded in the case of grinding *Miscanthus Giganteus* stems.

Using the same feeding flows and the same array of rotor speeds, if the mill was equipped with the 10 mm orifice sieve, the average material particles which pass through had values of 9.2 mm for one step hammers, 9.4 mm for two step hammers, 9.8 mm for three step hammers, and even 10 mm for triangle angle hammers.

In the case of grinding willow chips, grinded particle dimensions were significantly reduced, compared to the case of *Miscanthus Giganteus*. Thus, if using a 16 mm orifice sieve, the average dimensions of the grinded material are of 10.3 mm, both for one step hammers, as well as for two step hammers. If using a 10 mm orifice sieve, the average grinded material particles had values of 7.9 mm for the one step hammers, respectively 17.6 for two step hammers, but with a 5–15% higher energy consumption, which does not justify their use, more so since the two-step hammer wearing degree is greater.

The experimental results presented, both for *Miscanthus Giganteus* as well as for *Salix Viminalis*, the hammer type A (one step hammer) as being optimum, because the best biomass grinding degree, as well as the lowest specific energy consumption, were recorded in this scenario, in comparison to the other hammer types. Also, a uniform distribution of grinded material particle dimensions, disregarding rotor speed, was observed for the one step hammer type. This is the main reason for which the results for this hammer type were presented in the present paper. Moreover, we need to consider the fact that lignocellulosic biomass can have one of the three dimensions, much higher in comparison to the other two (smaller than the sieve orifices size. It is very probable that in the grinded material granulometric analysis, the same particle will not pass through the analyzer sieve orifices, so that they remain unpassed on one of the sieves, which can influence the results, even if the experiment is repeated two or three times.

## 4. Conclusions

The grinding process of a hammer mill is extremely difficult to optimize given the random nature of the material movement inside the milling chamber.

The distribution of the grinded material sizes resulted in a grinding process using a hammer mill is influenced by the dimension of the sieve orifices and the feeding flow with high influence on grinded material distribution, while the feeding flow presents a lower connection to the same distribution.

This paper presents a part of the studies done on the optimization of a hammer mill working process using statistical modelling based on experimental results. The process of optimizations continues with the statistical modelling optimization study of a hammer mill working process considering the energy consumption. The basis for the analysis conducted further are presented in this paper.

Given the complexity of the grinding process multiple studies must be realized in this field in order to generate highly efficient optimization process which can lead to better grind and a lower energy consumption.

The authors realized experimental determinations in more variants of equipping the hammer mill (regarding the hammer types and the evacuation sieve orifices) for an array of hammer mill rotor peripheric speeds, considered normal for different feeding flows. These input variables lead to very different results, but still the authors attempted to include them in a mathematical model which could sit at the base of better hammer mill design and exploitation.

The results also showed, that using hammers with multiple edges is not necessary, the best method found by the authors being the one step hammer type, both for grinding *Miscanthus Giganteus* biomass, as well as for grinding willow chips. The mathematical model used by the authors is built from a second–degree complex multi-criterial function, which is often described in appreciation of physical processes, especially in the field of agricultural or zoo-technical machine working processes.

Also, the method of equipping the hammer mill and its functional parameters must be interdependent with beneficiary requirements and future material destination.

The main conclusion, resulting from the presented experimental data, was that satisfying results could be obtained when using one step hammers, regarding grinded material particle dimensions and the specific energy consumption for both grinded material categories (*Miscanthus Giganteus* and *Salix Viminalis*). Moreover, through using the presented mathematical model in this paper, a correlation degree between the process parameters (hammer mill peripheral speed, feeding flow, mill sieve orifice dimensions, average grinded particle dimensions) of over 0.928 was obtained, in the cases of all four types of hammers.

Our study is a first attempt to optimize the parameters of the working regime of hammer mills used to crush plant, agricultural, and forestry biomass, based on several sets of experimental data. It can be the basis for the development and realization of other optimization models in the field.

**Author Contributions:** Conceptualization, G.P., G.V. and P.C.; Formal analysis, M.N.D. and P.T.; Funding acquisition, G.M. and M.C.; Investigation, G.M. and M.C.; Methodology, G.P. and G.V.; Software, P.C.; Supervision, G.P. and G.V.; Validation, P.C.; Writing—original draft, M.C., M.N.D. and P.T.; Writing—review and editing, G.M., M.N.D. and P.T. All authors have read and agreed to the published version of the manuscript.

**Funding:** This research was funded by the Sectoral Operational Programme Human Resources Development 2007–2013 of the Ministry of European Funds through the Financial Agreement POSDRU/159/1.5/S/137070; This work has been funded by the European Social Fund from the Sectoral Operational Programme Human Capital 2014–2020, through the Financial Agreement with the title "Scholarships for entrepreneurial education among doctoral students and postdoctoral researchers (Be Antreprenor!)", Contract no.51680/09.07.2019—SMIS code: 124539.

**Institutional Review Board Statement:** Not applicable.

**Informed Consent Statement:** Informed consent was obtained from all subjects involved in the study.

**Data Availability Statement:** Data presented in this manuscript are available upon request from the corresponding author.

**Conflicts of Interest:** The authors declare no conflict of interest.

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
