# Peer review of "Optimization Issues of a Hammer Mill Working Process Using Statistical Modelling"

_sustainability, doi:10.3390/su13020973_

Round 1

Reviewer 1 Report

Standardize the way numbers are presented in the tables. Change commas to dots.

Author Response

Dear reviewer,

Thank you for the review of our paper, we have done the suggestions made meaning:

- In table 1 and 2 we changed commas to dots

- English language was corrected accordingly.

Thank you!

Best regards,

Gheorghe Voicu

Reviewer 2 Report

The authors deal with an interesting subject, which can be suitable for this journal. Having said that, there are some remarks that should be solved, like the following.

  • The text has to be revised in order to avoid some typos or sentences that do not make much sense or are difficult to understand. For instance, the first sentence of the abstract is long and should be simplified.
  • Keywords: energetic biomass?
  • Introduction: You should add more references concerning the different kinds of lignocellulosic biomass, including, at least, the following references:
    • https://doi.org/10.3390/ma8041413
    • https://doi.org/10.1016/j.biombioe.2020.105888
  • Materials and Methods:
    • Section 2.1. date or data?
    • These scientific terms should be Miscanthus Giganteus and Salix Viminalis (both with capital letters), right? Unify it throughout the text.
    • Tables 1 and 2: Replace “÷” for “-“ between the different particle sizes.
  • Results and discussion:
    • All tables or figures with a numerical reference should be in capital letters (e. g., Figure 2, Table 1, etc.).
    • Figures 4 and 5. If there were no experiments with 7 mm of sieve diameter, you should remove it from these figures, maybe.
    • Figure 6. If the experiments were the same for the same sieve diameter, why not use the average and error bars? Or at least you can differentiate between the same sieve diameter by pointing out the material flow.
    • Avoid the introduction term “in this paragraph”. It is better “in this section”, for instance.
  • Conclusions: You should add some specific results of your study, the ones you consider to be the most interesting.

Author Response

Dear reviewer,

Happy New Year and thank you for the review of our paper, we have done the suggestions as follows:

  1. The text has to be revised in order to avoid some typos or sentences that do not make much sense or are difficult to understand. For instance, the first sentence of the abstract is long and should be simplified.

Abstract

- the first sentence in the abstract was rewritten.

  1. Keywords: energetic biomass?

- the keyword “energetic biomass” was rewritten for a better understanding as “energetic crop biomass”,

  1. Introduction: You should add more references concerning the different kinds of lignocellulosic biomass, including, at least, the following references:

https://doi.org/10.3390/ma8041413

https://doi.org/10.1016/j.biombioe.2020.105888

Introduction

- in the Introduction the references mentioned were included as well as other2 that present the different kinds of lignocellulosic biomass;

  1. Materials and Methods:
    1. Section 2.1. date or data?
    2. These scientific terms should be Miscanthus Giganteus and Salix Viminalis (both with capital letters), right? Unify it throughout the text.
    3. Tables 1 and 2: Replace “÷” for “-“ between the different particle sizes.

Material and Methods

- in Section 2.1 “data” was replaced with “results” for a better understanding.

- the terms Miscanthus Giganteus and Salix Viminalis were unified throughout the text.

- in Tables 1 and 2 we replaced “÷” for “-“between the different particle sizes.

  1. Results and discussion:
  • All tables or figures with a numerical reference should be in capital letters (e. g., Figure 2, Table 1, etc.).
  • Figures 4 and 5. If there were no experiments with 7 mm of sieve diameter, you should remove it from these figures, maybe.
  • Figure 6. If the experiments were the same for the same sieve diameter, why not use the average and error bars? Or at least you can differentiate between the same sieve diameter by pointing out the material flow.
  • Avoid the introduction term “in this paragraph”. It is better “in this section”, for instance.

 Results and discussions

- all tables or figures with a numerical reference were put in capital letters

- 7 mm of sieve diameter was removed from Figures 4 and 5;

- Additional information was added by also presenting the average values along with the error bars.

- the syntax “in this paragraph” was replaced with “in this section”;

- English language was corrected accordingly.

  1. Conclusions: You should add some specific results of your study, the ones you consider to be the most interesting.

Conclusions

Additional information was added along the paper as well as a few more conclusions that could be drawn through the experiments and the analysis done.

Thank you once again for the review and we have to mention that we added some aspects/information in the paper for a better understanding of the analysis and the results presented.

Best regards,

Gheorghe Voicu

Reviewer 3 Report

Dear Authors,

I consider that your paper provides sufficient analysis and insight to be an interesting and practical addition to the field and can help design ideal hammer mills depending on wanted quality of ground stems.

Overall some questions were raised from reading your manuscript:

1) You explored a range of frequencies from 40 to 50Hz, yet that it was an important parameter (demonstrated in Figure7). Why did you not consider higher or smaller frequencies from that range (was it due to a hardware limitation?)

2)Why did you stop your interpolation polynom to a second degree based on the fact that it did not provide fully satisfactory results? (equation 2)

3)Figure 7 presents in only one case (although the tables allow for plotting other data). This demonstrates the impact of sieve hole diameter and rotor frequency. To improve the presentation, I think you are missing the opportunity to showcase your model results by plotting only one hammer configuration (for which additional data is presented in table 7). There should be a plot with the different hammer types in this case or if you believe that based on the small correlation difference, changing the hammer type does not make such a difference.

4)Based on your study I could not find a sentence stating what are the optimal number of edges in the hammers

Best regards and best wishes for 2021 

Author Response

Dear reviewer,

Happy New Year and thank you for the review of our paper. Regarding the questions raised from reading the manuscript we must mention the following:

1) You explored a range of frequencies from 40 to 50Hz, yet that it was an important parameter (demonstrated in Figure7). Why did you not consider higher or smaller frequencies from that range (was it due to a hardware limitation?)

Answer: From the most relevant specialty literature papers we drew the conclusion that the most practical rotor frequency is 50 Hz. Following the grinding process the optimal particle size dimension for biomass was observed to be obtained at this frequency, thus this is the main reason for us choosing frequencies from 40 to 50Hz.

2) Why did you stop your interpolation polynom to a second degree based on the fact that it did not provide fully satisfactory results? (equation 2)

Answer: In general, for the mathematical expression of physical processes, polynomial functions higher than the second degree are not used, especially when more than two independent variables that characterize the process are considered.

3) Figure 7 presents in only one case (although the tables allow for plotting other data). This demonstrates the impact of sieve hole diameter and rotor frequency. To improve the presentation, I think you are missing the opportunity to showcase your model results by plotting only one hammer configuration (for which additional data is presented in table 7). There should be a plot with the different hammer types in this case or if you believe that based on the small correlation difference, changing the hammer type does not make such a difference.

4) Based on your study I could not find a sentence stating what are the optimal number of edges in the hammers.

Answer 3 and 4: It is presented only one case (Figure 7) because out of our observations and results obtained, we could see that one step hammer gave the best results for the specific energy consumption as well as for the quality of the grinded material. We presented this information in the paper also.

Thank you once again for the review and we must mention that we added some aspects/information in the paper for a better understanding of the analysis and the results presented.

Best regards,

Gheorghe Voicu